# TrajCLIP: Pedestrian Trajectory Prediction Method Using Contrastive Learning and Idempotent Networks

**Pengfei Yao**[1]
yaopengfei22@mails.ucas.ac.cn

**Yinglong Zhu**[2♯]
z.yinglong@bupt.edu.cn

**Huikun Bi**[3]
bihuikun@senseauto.com

**Tianlu Mao**[4**]
ltm@ict.ac.cn

**Zhaoqi Wang**[4]
zqwang@ict.ac.cn

[1]Institute of Computing Technology, University of Chinese Academy of Sciences
[2]Beijing University of Posts and Telecommunications
[3]SenseTime
[4]Beijing Key Laboratory of Mobile Computing and Pervasive Device

## Abstract

The distribution of pedestrian trajectories is highly complex and influenced by the scene, nearby pedestrians, and subjective intentions. This complexity presents challenges for modeling and generalizing trajectory prediction. Previous methods modeled the feature space of future trajectories based on the high-dimensional feature space of historical trajectories, but this approach is suboptimal because it overlooks the similarity between historical and future trajectories. Our proposed method, TrajCLIP, utilizes contrastive learning and idempotent generative networks to address this issue. By pairing historical and future trajectories and applying contrastive learning on the encoded feature space, we enforce same-space consistency constraints. To manage complex distributions, we use idempotent loss and tightness loss to control over-expansion in the latent space. Additionally, we have developed a trajectory interpolation algorithm and synthetic trajectory data to enhance model capacity and improve generalization. Experimental results on public datasets demonstrate that TrajCLIP achieves state-of-the-art performance and excels in scene-to-scene transfer, few-shot transfer, and online learning tasks.

## 1 Introduction

Pedestrian trajectory prediction involves predicting future paths based on observed historical trajectories. This task is crucial in real-world applications such as autonomous driving [4, 13], and robotics [11]. It is challenging to model diverse trajectory distributions across different scenes (e.g., roundabouts and intersections), various patterns (e.g., staggering and strolling), and different interactions (e.g., walking together or avoiding strangers) due to their complexity.

The existing methods [31, 21, 6] assume that the training and test motions follow the same distribution to effectively model complex distributions within a specific range. These approaches focus on modeling the high-dimensional feature space distribution of trajectories within a particular dataset. They use generative models, such as Conditional Variational AutoEncoder [23, 24] and diffusion, to change the feature space of past trajectories into the feature space of future trajectories. This process aids in generating predicted trajectories and fitting the distribution of pedestrian trajectories to specific datasets. However, as shown in Fig. 1, this approach has limitations. It involves reducing

---

*⋆ denotes the corresponding author. ♯ contributed equally to this work as co-first author.

38th Conference on Neural Information Processing Systems (NeurIPS 2024).

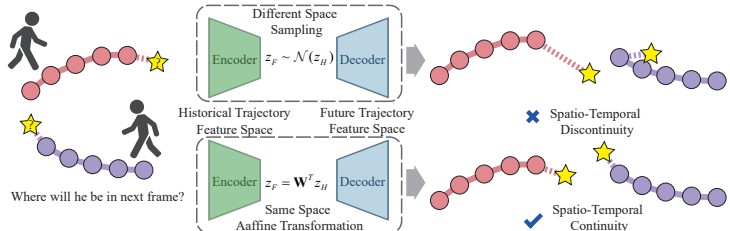

Figure 1: The existing methods use an encoder to sample from a normal distribution in order to generate predicted trajectories. However, this approach leads to different feature spaces for historical and future trajectories, causing problems during sampling, such as sharp turns or predictions of positions far exceeding normal speeds. Our method ensures consistency between the two feature spaces, enabling position prediction through affine transformations and preserving spatio-temporal continuity.

the feature space to match the distribution of a specific dataset to improve prediction accuracy. Nonetheless, this projection operation creates a gap in the feature space between the historical and future trajectories, limiting the model's ability to make predictions across different scenarios and to adapt to new scenarios.

We believe that predicting future trajectories in trajectory prediction tasks is significantly different from tasks such as classification [32] and tracking [30], where the input and output are inconsistent. In trajectory prediction, the feature space of the input and output remains consistent. Therefore, mapping historical trajectory features to future trajectory features in the same feature space can effectively bridge the feature space gap and model complex feature space distribution more accurately.

We have developed a method called TrajCLIP for predicting trajectories, which is based on contrastive learning and idempotent neural networks. This method was inspired by the approach used in the paper by [18], which involves limiting the feature space of heterogeneous data. To achieve this, we created a trajectory feature encoder that utilizes frequency domain additivity and Fourier transforms reversibility to combine trajectory Interaction Features (STIF) in the time domain and Interaction Features (SAIF) between agents in the frequency domain. We utilized cross-entropy loss for constructive learning between historical trajectory and future trajectory encoders to ensure that the feature spaces of the two encoders are the same. Subsequently, we employed an idempotent generation network (IGN) as a global feature mapping framework to map the same set of features from past trajectory features to future trajectory features. Additionally, we constrained the growth of the specific distribution to allow for more general usage. Finally, the mapped future trajectory features were decoded into predicted trajectories. We also introduced a neighbor-based interpolation algorithm and used a trajectory generation model to create a synthetic trajectory to enhance the model's ability to describe complex distributions. Furthermore, we trained a lightweight version (TrajCLIP-tiny) without these augmentations. We conducted experimental comparisons with existing trajectory prediction methods on various tasks and datasets. The experiments demonstrated that our method achieves state-of-the-art accuracy and outperforms existing methods in cross-scene generalization, small sample transfer, and online tasks.

The contributions of our work are summarized as follows: (1) In this study, TrajCLIP is presented as a proposed method. The model effectively represents and generalizes complex feature distributions by integrating the feature spaces of historical and future trajectories. An idempotent neural network with global feature mapping capabilities is utilized, enabling strong cross-scene generalization abilities. (2) We proposed a novel trajectory feature encoder combines features from both the time domain and the frequency domain using STIF and SAIF to model pedestrian trajectories. (3) We have created a novel method for measuring trajectory similarity, which we use as a label for contrastive learning. This helps to narrow down the feature space of past and future trajectories. To our knowledge, this is the first application of contrastive learning to trajectory prediction. This is the first work that considers the feature space alignment between the history and future trajectory. Our experiments demonstrate that TrajCLIP performs exceptionally well across different tasks and datasets.

## 2 Related Work

### 2.1 Pedestrian Trajectory Prediction

Pedestrian trajectory prediction is approached as a sequence-to-sequence challenge. Social-LSTM [1] first uses LSTM to predict trajectories deterministically. Social-GAN [7] uses a Generative Adversar-

ial Network (GAN) to make many possible future predictions for an input trajectory while taking into account the inherent multimodality of human motion. Trajectron++ [21] takes in motion trajectories and heterogeneous data, and it handles multi-modality by using CVAE to sample latent variables in high-dimensional space. AgentFormer [31] proposes agent-aware attention to simultaneously capture the spatiotemporal features of trajectories and improves the sampling method of the CVAE framework to model more intricate trajectory modalities. MID [6] was the first to use diffusion, which gradually gets rid of uncertainty in all walkable areas by using a high-dimensional denoiser until the desired path is reached. A new model for predicting trajectories called Flow-Chain [14] uses normalizing flow to quickly figure out the probability distribution for each point on the trajectory. TUTR [22] employs a unified transformer to model trajectories, directly providing the predicted trajectory and the probability of each mode, which effectively eliminates the need for post-processing. Long-tail analysis [15] uses Kalman filter to directly predict output trajectory errors as a basis, distinguishing difficult and easy samples. It then conducts contrastive learning on difficult (long-tail part) historical trajectories to enhance the feature representation capabilities of the long-tail part of the dataset. FEND [25] uses VAE to extract features from the entire trajectory and performs offline clustering on the features to obtain labels; during training, it uses offline labels for contrastive learning on historical trajectories to guide the feature generation of historical trajectories and enhancing the prediction of the long-tail part. However, projecting past trajectory features into separate feature spaces creates a feature space gap that restricts the model's capacity to generalize and accurately represent complex trajectory distributions.

## 2.2 Generative Framework

Compared with the deterministic framework, the generative framework focuses on modeling data distribution in the dataset and is well-suited for tasks involving one input and multiple outputs. GAN [5] introduces a generative adversarial network, allowing the generator to model the real data distribution through adversarial learning with the discriminator. VAE [12] introduces latent variables, uses the encoder to construct the latent feature space representation of the data, and uses the decoder to restore and reconstruct the feature tensor. CVAE [24] adds a control signal on top of VAE [12]. This signal changes the variational lower bound in the conditional probability state, which lets the model control the generated data distribution based on the signal that goes into it. Diffusion [9] models the data distribution by simulating the diffusion process in high-dimensional space. Adding high-dimensional denoisers gradually lowers uncertainty about the data distribution, leading to samples that are similar to the training data.

In these methods, there is a separation between the feature spaces of the control signal and reconstructed data. However, future and historical trajectory features should be in the same feature space, and modeling these independently will lead to a natural gap. IGN [23] aims to model global feature mapping, which constrains the distribution of specific datasets on the expected manifold through idempotent loss and compaction loss. This approach prevents the model from being overfitted to the distribution of specific datasets and enables data generation within the same feature space. DALLE [19] uses CLIP [18] to align the different feature spaces of text and images. It then uses the encoded feature vector of the text as a control signal to make different kinds of images using GLIDE [17]. We utilize contrastive learning to ensure consistency between the feature spaces of historical trajectory and future trajectory, based on the concept of aligning feature spaces. Subsequently, we employ IGN mapping to generate trajectories.

## 3 Methodology

### 3.1 Problem Formulation

We generalize the trajectory prediction task as follows. Given a scenario comprising $N$ agents, the historical information for each agent includes the two-dimensional coordinate position $p_t^n = (x_t^n, y_t^n)$ of agent $n$ at time $t$. The trajectory prediction task is to predict future position sequence $\hat{p}_{t_0+1 \sim t_0+T_f}^n$ based on the observed historical agent position sequence $p_{t_0-T_h+1 \sim t_0}^n$.

### 3.2 Overall Framework

We propose a trajectory prediction method called TrajCLIP, which uses contrastive learning and and idempotent generative network, as illustrated in Fig. 2. Our TrajCLIP comprises four parts: (1). A trajectory encoder that models interactions between agents, scene-agent interactions in the

frequency domain, and trajectory spatiotemporal characteristics in the time domain. (2). A CLIP pairing learning module that uses cross-entropy loss to maintain feature space consistency between past and future trajectories. (3). A feature manifold predictor that employs idempotent and tightness loss to train an IGN for mapping past trajectory features into future trajectory features in the same feature space. (4). A trajectory manifold decoder that uses a pre-trained task decoder from the trajectory encoder to generate predicted trajectories. We will provide detailed introductions to each part in the subsequent sections. Specific implementation details can be found in Sec. 4.2.

### 3.3 Data Augmentation

Various data augmentation techniques are used in current methods to enhance the performance of pedestrian trajectory prediction models, including rotation for robustness, scale alignment for training simplicity, and trajectory origin alignment for addressing absolute position ambiguity. In our research, we incorporate minor perturbations (as discussed in [20]) into trajectories to generate more reliable results. We achieve this by simulating scenarios using available datasets, as demonstrated in the study conducted by [33]. This approach allows us to enhance the existing dataset. Additionally, we have developed an interpolation technique that increases the model capacity by integrating neighboring trajectory sample points.

We formulate the adjacent point interpolation algorithm as illustrated in Fig. 4. If the vectors $\overrightarrow{p_{t-1}p_t}$ and $\overrightarrow{p_{t+1}p_{t+2}}$ are on the same side of $\overrightarrow{p_t p_{t+1}}$ for four consecutive positions $p_{t-1 \sim t+2}$, we can calculate the acute angle between the two adjacent trajectories. To derive the interpolated trajectory, the data is divided into three equal segments. Then, we find the intersection point of the third line and connect them. When the vectors are on opposite sides or at the beginning and end of the sequence, the trisect points are chosen as the interpolation points. In the rest of this paper, we will use the symbol $p^n_{t-T_h+1 \sim t}$ to represent the interpolated trajectory, for simplicity. Despite the potential decrease in accuracy, the generated and interpolated data can still be used to improve the model's capabilities and capture more complex distributions, considering the multimodal nature of the pedestrian trajectory prediction task.

### 3.4 Trajectory Encoder

In our approach, we use two structurally identical encoders, as shown in Fig. 3. These encoders represent the historical trajectory $H$ and the future trajectory $F$. We utilize the agent-aware attention $\varphi_{Att}$ from Agentformer [31] to extract the STIF, and employ a transformer based on Fourier Transform to extract the SAIF. By fusing these features, we train both encoders to decode future trajectories. Additionally, the historical trajectory encoder aims to accurately predict the subsequent frame position to maintain spatio-temporal continuity.

We define the time-series set $\mathcal{T}_H = \{t_0 - T_h + 1, ..., t_0 - 1, t_0\}$ for the historical trajectory encoder, and $\mathcal{T}_F = \{t_0 - 1, t_0, t_0 + 1, ..., t_0 + T_f\}$ for future trajectory encoders. To obtain the spatio-temporal

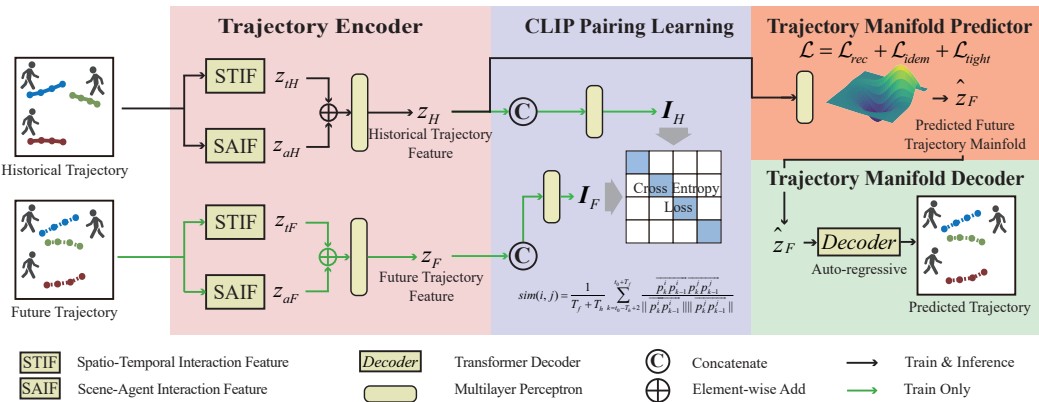

Figure 2: Overall framework of our proposed TrajCLIP. Introduction to each part can be found in Section 3.2, and specific implementation details can be found in Sec. 4.2.

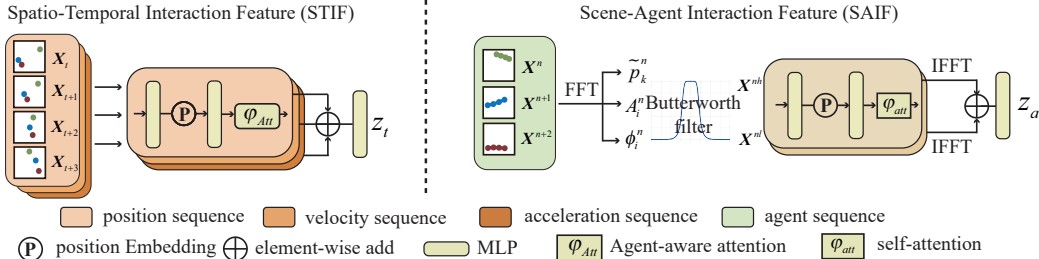

Figure 3: The left side shows the Architecture of Spatio-Temporal Interaction Feature, while the right side shows the Scene-Agent Interaction Feature.

interaction feature (STIF), we arrange the trajectories in a scene into $\mathbf{X}_t = ((p_t^n \mid n = 1, ..., N) \mid t \in \mathcal{T})$, where time is used as the index. At each time step, we use a straightforward linear mapping to tokenize the $X$ matrix and include positional encoding to maintain the temporal information of the trajectory token sequence. We employ agent-aware attention to extract both the temporal features of each individual trajectory and the interaction features between different trajectories simultaneously. The feature is denoted as $z_t$ and can be formally described as follows:

$$f : \mathbf{M} \to \varphi_{Att}(\mathbf{W}_2(\mathbf{W}_1^T \mathbf{M} \oplus \Gamma)) \quad z_t = MLP[f(\mathbf{X}_t) + f(\mathbf{V}_t) + f(\mathbf{A}_t)] \tag{1}$$

where $\mathbf{W}_1^T$ and $\mathbf{W}_2$ represent trainable matrices, $\varphi_{Att}$ denotes agent-aware attention, $\Gamma$ denotes sine and cosine position encoding [31], and $MLP$ denotes a multi-layer perceptron. We can easily derive the velocity $\mathbf{V}_t$ and acceleration $\mathbf{A}_t$ of the sequence using $\mathbf{X}_t$. Similarly, we employ equation 1 to extract features for both of these sequences. Parameters are not shared among different sequences.

The Scene-Agent Interaction Feature (SAIF) characterizes the shared representation of the scene by the agents. We believe that the interactions between agents and the scene are implicit and shared. These interaction features should be represented uniformly, without including the individual characteristics of the agents. As there is no direct way to merge the agents' trajectories in the time domain, we leverage the additivity in the frequency domain to eliminate the fine-grained features of the agents' trajectories, thereby emphasizing the interaction information of the scene.

We organize the trajectories in a scene into $\mathbf{X}^n = ((p_t^n \mid t \in \mathcal{T}) \mid n = 1, ..., N)$ with agent as the index. We apply a Fast Fourier Transform (FFT) to each agent's trajectory based on Equation 2. Utilizing conjugate symmetry, we can derive the cosine expression for both amplitude and phase.

$$\tilde{p}_k^n = \sum_{i \in \mathcal{T}}^t p_i^n e^{-j\frac{2\pi}{T_h}tk} = \sum_{i \in \mathcal{T}}^t A_i^n cos(\frac{2k\pi}{T_h} + \phi_i^n) \tag{2}$$

We employ a Butterworth filter to separate high-frequency information ($\tilde{p}_k^n$, $A_i^n$, $\phi_i^n$) to enhance modeling efficiency. Then we superimpose and concatenate the low-frequency signals $\mathbf{X}^{nl}$ and high-frequency signals $\mathbf{X}^{nh}$ of all agents in the scene respectively, and employ self-attention to extract feature.

$$\tilde{\mathbf{X}}_n^g = \mathbf{W}_4(\mathbf{W}_3^T \mathbf{X}^{ng} \oplus \Gamma), g \in \{h, l\} \quad z_a = MLP[IFFT(\varphi_{att}^h(\tilde{\mathbf{X}}^{nh}) + \varphi_{att}^l(\tilde{\mathbf{X}}^{nl}))] \tag{3}$$

where $\mathbf{W}_3$ and $\mathbf{W}_4$ denote trainable matrices, $\varphi_{att}^g$ denotes multi-head self-attention, and $IFFT$ denotes Inverse Fast Fourier Transform (IFFT). IFFT is a lossless and reversible transformation, so the extracted features can be inversely transformed into time-domain features without any information loss and fused with STIF in the same feature space. The final output of the trajectory encoder is the result of a linear transformation of the sum of $z_t$ and $z_a$.

## 3.5 CLIP Pairing Learning Module

We aim to transform the historical trajectory features into estimated future trajectory features, displaying temporal continuity to aid in trajectory prediction. It is important to maintain consistency between the feature spaces of past and future trajectories to satisfy spatio-temporal constraints. Therefore, it is necessary to retrain the historical trajectory encoder to align its feature space with that of the future trajectory encoder. We utilize CLIP [18], a method for aligning feature spaces of different types of information representation, such as text and images, using contrastive learning. We apply this concept to ensure consistency between the feature spaces in the past and future. The

specific algorithm is outlined in Algorithm 1. The similarity matrix within each training batch of trajectories is calculated using Equation 4. The mean included angle cosine of the velocity direction vectors between trajectories $i, j \in \{1, 2, ..., batch\_size\}$ is computed at each corresponding sample point in both the historical and future data.

$$sim(i,j) = \frac{1}{T_f + T_h} \sum_{k=t_0-T_h+2}^{t_0+T_f} \frac{\overrightarrow{p_k^i p_{k-1}^i} \cdot \overrightarrow{p_k^j p_{k-1}^j}}{||\overrightarrow{p_k^i p_{k-1}^i}|| \, ||\overrightarrow{p_k^j p_{k-1}^j}||} \quad (4)$$

Simultaneously, we calculate features $z_H^i$ and $z_F^i$ for each agent using the historical trajectory encoder and future trajectory encoder, and then combine the corresponding encoder features. Next, we align the dimensions using a trainable matrix $\mathbf{W}$. Finally, we use the dot product to obtain the similarity scores and optimize them using the cross-entropy loss with the similarity matrix. Contrastive learning ensures that the feature space from the two encoders is consistent, which makes subsequent feature space mapping possible.

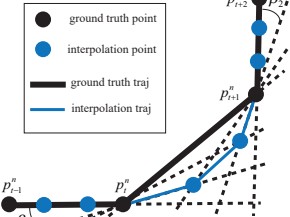

Figure 4: Schematic diagram of Adjacent Point Interpolation.

---

**Algorithm 1** CLIP Pairing Learning

**Input**: agent index in batch: i=1, ..., batch_size

1: Calculate similarity matrix $\mathbf{S}$ using equation.
2: $z_H^i \leftarrow MLP[z_{aH}^i + z_{tH}^i]$ for i in batch.
3: $z_F^i \leftarrow MLP[z_{aF}^i + z_{tF}^i]$ for i in batch.
4: $\mathbf{Z}_H \leftarrow concate([z_H^i$ for i in batch])
5: $\mathbf{Z}_F \leftarrow concate([z_F^i$ for i in batch])
6: $\mathbf{I}_H \leftarrow normalize(\mathbf{W}_H \mathbf{Z}_H, axis = 1)$
7: $\mathbf{I}_F \leftarrow normalize(\mathbf{W}_F \mathbf{Z}_F, axis = 1)$
8: $\mathcal{L}_H \leftarrow cross\_entropy\_loss(\mathbf{S}, \mathbf{I}_H \mathbf{I}_F^T, axis = 0)$
9: $\mathcal{L}_F \leftarrow cross\_entropy\_loss(\mathbf{S}, \mathbf{I}_H \mathbf{I}_F^T, axis = 1)$
10: backpropagation with $\mathcal{L}_{clip} \leftarrow (\mathcal{L}_H + \mathcal{L}_F)/2$

---

### 3.6 Trajectory Manifold Predictor

Following the CLIP pairing learning module, the historical and future trajectory embeddings are constrained to share the same feature space. Future trajectory features can be derived by applying a linear transformation to the historical trajectory features. Because there is inherent uncertainty surrounding the intentions of agents, the model's projected future trajectory should be represented as a probability distribution of the agents' positions. Therefore, it is necessary to estimate a manifold for the distribution of future trajectory features.

We have noticed that there is an idempotent phenomenon in this consistent feature space. Specifically, the training instances should have fixed points in the feature space as their future trajectories. Our goal is to create an estimated manifold within which the future historical trajectory will be located. This statement aligns with the fact that the distribution of predicted future trajectories, sampled in real space, includes the original trajectory. Therefore, the loss function used to train the feature transformation network should encompass both the reconstruction loss and the idempotent loss, as shown in the equation.

$$\widehat{\mathbf{Z}}_F = MLP[\mathbf{Z}_H; \epsilon], \ \epsilon \sim \mathcal{N}(0,1)$$
$$\mathcal{L}_{rec} = ||\widehat{\mathbf{Z}}_F - \mathbf{Z}_F||_2 \quad (5)$$
$$\mathcal{L}_{idem} = ||MLP_{\theta'}[MLP_\theta[\mathbf{Z}_F]] - MLP_\theta[\mathbf{Z}_F]||_2$$

We additionally incorporate Gaussian noise sampling into the input to model the agents' intentions. Additionally, since the feature space is subject to consistency constraints, trajectories from multiple source domains correspond to different manifolds within this space. If, on the other hand, the model runs into complex distributions, the idempotent loss can cause representation collapse, also known as overfitting [23]. To mitigate this, a tightness loss is also needed to constrain the manifold expansion.

$$\mathcal{L}_{tight} = -||MLP_\theta[MLP_{\theta'}[\mathbf{Z}_F]] - MLP_{\theta'}[\mathbf{Z}_F]||_2 \quad (6)$$

where $\theta$ denotes the MLP parameter, and $\theta'$ denotes freezing the MLP parameter at training. The detailed reason and approach for adopting this training strategy can be found in the [23].

### 3.7 Trajectory Manifold Decoder

We employ an auto-regressive decoder with the standard Transformer architecture. No additional training tasks are required at this stage. During the trajectory encoding phase, both the historical

trajectory feature encoder and the future trajectory feature encoder aim to output position coordinates for the period from $t-1$ to $t+T_f$ . Two additional frames of historical trajectory are included to better model spatiotemporal continuity. The former is similar to a prediction task, while the latter resembles a reconstruction task. The future trajectory feature decoder, trained during the encoder phase, is directly used for trajectory decoding at this stage.

$$z = MLP(z_a + z_t)$$
$$\hat{p}^n_{t-1\sim t+T_f} = Decoder(z)$$
$$\mathcal{L} = \frac{1}{N \times (T_f + 2)} \sum_{n=1}^{N} \sum_{i=t-1}^{t+T_f} ||\hat{p}^n_i - p^n_i||_2 \tag{7}$$

## 4 Experiments

### 4.1 Datasets And Metrics

**Datasets.** We evaluate our method using three popular trajectory prediction datasets: These include the ETH-UCY dataset, with its five scenes named ETH, HOTEL, UNIV, ZARA1, and ZARA2, which is considered the standard for predicting the paths of pedestrians. Additionally, we used the SDD dataset, the first large-scale dataset that collects agent trajectories in eight different outdoor scenes, and the SNU dataset, which includes different patterns such as walking straight, strolling, and staggering in indoor scenes.

**Metrics.** We use a standard evaluation approach for pedestrian trajectory prediction. We observe 8 frames of trajectories to predict 12 frames of future trajectories. Each time, we sample 20 predicted trajectories and compare the prediction results closest to the ground truth [31]. Our evaluation metrics of choice are Average Displacement Error (ADE) and Final Displacement Error (FDE).

$$ADE = \frac{1}{N \times T_f} \sum_{n=1}^{N} \sum_{i=t+1}^{t+T_f} ||\hat{p}^n_i - p^n_i||_2 \quad FDE = \frac{1}{N} \sum_{n=1}^{N} ||\hat{p}^n_{t+T_f} - p^n_{t+T_f}||_2 \tag{8}$$

### 4.2 Implementation Detail

For training process, our batch size is set to 64, epochs to 100, and the learning rate is 0.01, which is half-formed every 25 epochs. We use the Adam optimizer. Our model is trained on an RTX 3090, with the encoder and contrastive learning pre-training requiring approximately 8 GPU hours, and the full framework training taking about 9.6 GPU hours. Our method is divided into three stages: offline training, inference, and online adaptation.

**Offline Training.** First, we apply data augmentation to process trajectory data. Next, we pre-train historical and future trajectory encoders. Once these two encoders have reached convergence in their training, we use contrastive learning to impose constraints on their feature spaces for spatio-temporal constraints. During contrastive learning, we fine-tune the future trajectory decoders using pre-training tasks. It's important to note that during this step, the backpropagation gradient from the pre-train task is truncated only in the decoder section to adapt the future trajectory decoder to changes in the future trajectory encoder and maintain its decoding capabilities. Finally, we train the trajectory manifold predictor. The gradient for this training task is also truncated, meaning the parameters of the encoder are no longer updated.

**Inference.** During the inference stage, we do not engage in comparative learning or future trajectory encoding, unlike the offline training stage. Instead, we conduct interpolation and alignment on the input historical pedestrian trajectory that needs to be predicted. We then utilize the historical trajectory encoder and trajectory manifold predictor to extract future trajectory features. Finally, we use the trajectory manifold decoder to generate the predicted trajectory.

**Online Adaption.** During the online adaptation stage, we use a lightweight version of our model (TrajCLIP-tiny) for real-time inference. We freeze the trajectory encoder and manifold decoder and only fine-tune the parameters of the updated trajectory manifold predictor with a small learning rate based on newly observed trajectories. By adjusting the mapping between historical trajectory and future trajectory, the manifold predictor can map the same historical trajectories into future trajectories adapted to the current environment.

### 4.3 Experiment and Analysis

We conduct comparative experiments with state-of-the-art methods on four common trajectory prediction tasks: standard trajectory prediction, scene-to-scene transfer, few-shot transfer, and online adaptation. These experiments provide a comprehensive evaluation of our method's prediction

| | ETH | HOTEL | UNIV | ZARA1 | ZARA2 | AVG | SDD | SNU |
|---|---|---|---|---|---|---|---|---|
| Trajectron++ | 0.61/1.02 | 0.19/0.28 | 0.30/0.54 | 0.24/0.42 | 0.18/0.31 | 0.30/0.51 | 10.65/16.74 | 0.35/0.50 |
| AgentFormer | 0.45/0.75 | 0.14/0.22 | 0.25/0.45 | 0.18/0.30 | 0.14/0.24 | 0.23/0.39 | 8.24/15.21 | 0.32/0.46 |
| MID | 0.39/0.66 | 0.13/0.22 | 0.22/0.45 | 0.17/0.30 | 0.13/0.27 | 0.21/0.38 | 7.61/14.32 | 0.30/0.45 |
| GP-Graph | 0.43/0.63 | 0.18/0.30 | 0.24/0.42 | 0.17/0.31 | 0.15/0.29 | 0.23/0.39 | 9.10/13.76 | 0.31/0.47 |
| EqMotion | 0.40/0.61 | 0.12/0.18 | 0.23/0.43 | 0.18/0.32 | 0.13/0.23 | 0.21/0.35 | 7.52/12.88 | 0.28/0.42 |
| TUTR | 0.40/0.61 | 0.11/0.18 | 0.23/0.42 | 0.18/0.34 | 0.13/0.25 | 0.21/0.36 | 7.76/12.69 | 0.29/0.43 |
| SingularTrajectory | 0.35/0.42 | 0.13/0.19 | 0.25/0.44 | 0.19/0.32 | 0.15/0.25 | 0.21/0.32 | 7.26/12.58 | 0.29/0.41 |
| Ours-tiny | 0.39/0.62 | 0.13/0.19 | 0.22/0.42 | 0.19/0.32 | 0.13/0.24 | 0.21/0.36 | 7.04/13.40 | 0.29/0.42 |
| Ours(TrajCLIP) | 0.36/0.57 | 0.10/0.17 | 0.19/0.41 | 0.16/0.28 | 0.11/0.20 | 0.18/0.33 | 6.29/11.79 | 0.25/0.39 |

Table 1: Prediction performance on the ETH-UCY, SDD, SNU dataset using ADE/FDE. We sample 20 possible trajectories and calculate metrics based on the trajectory that is closest to the ground truth. The bold/underlined font denotes the best/second-best result.

accuracy, generalization ability across different scenarios, learning capability with few samples, and online learning ability in dynamic environments. Additionally, we perform ablation studies on our proposed data augmentation method and various modules to verify the effectiveness of our approach.

**Standard Trajectory Prediction Task**. Follows the standard trajectory prediction setting, we compare our model, TrajCLIP with six recent models with the best performance (1) Traj++ [21], (2) AgentFormer [31], (3) MID [6], (4) GP-Graph [2], (5) EqMotion [27] (6) TUTR [22] and (7) SingularTrajectory [3] on the three datasets ETH-UCY, SDD, and SNU in Table. 1. From the table, our TrajCLIP model outperforms all other models, achieving the smallest ADE/FDE value and reducing the prediction error by up to 17%. Additionally, we develop TrajCLIP-tiny, which eliminates trajectory interpolation and reduces the attention matrix in the model, resulting in a 76.92% reduction in parameters. This allows TrajCLIP-tiny to adapt to various trade-offs between computational cost and prediction performance in real-world application scenarios. Moreover, TrajCLIP-tiny demonstrates competitive or superior performance in most of the dataset evaluation scenes.

**Scene-to-Scene Transfer Task**. In this experiment, our goal is to assess the model's ability to generalize across different scenes by training and evaluating it on different scenes within the ETH-UCY dataset. Following the approach in [29], we train the model on one scene and test it on another. To ensure a fair comparison with existing baselines in this new setup, we train the baselines using the training data from the source scene and the validation data from the target scene. For example, taking column A2B from right part of Table 2, we train the baselines using the training data from dataset A and the validation data from dataset B, and then evaluate the model's performance using ADE/FDE on the test data from dataset B. It's important to note that the test and evaluation sets are independent, so the model only has access to trajectories from the validation set during evaluation.

We compare them with the existing trajectory prediction methods: the transfer learning work T-GNN [29] based on domain adaptation, the meta-learning work K0 [10] using ALPaCA [8] adaptation, and tra2tra [28] which performs better in transfer learning. The experiment result is shown in right part of Table 2. We discover that TrajCLIP-tiny exhibits competitive performance in the majority of scenes thanks to strong generalization capability from manifold predictor. In certain situations, it even outperforms trajectory prediction methods specifically designed for transfer learning. Moreover, TrajCLIP has achieved optimal or sub-optimal accuracy in the five scene-to-scene transfer tasks.

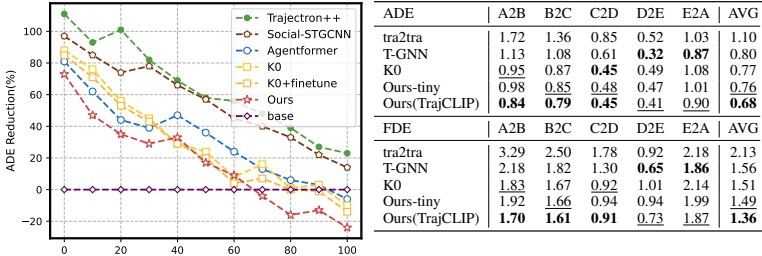

| ADE | A2B | B2C | C2D | D2E | E2A | AVG |
|---|---|---|---|---|---|---|
| tra2tra | 1.72 | 1.36 | 0.85 | 0.52 | 1.03 | 1.10 |
| T-GNN | 1.13 | 1.08 | 0.61 | 0.32 | 0.87 | 0.80 |
| K0 | 0.95 | 0.87 | 0.45 | 0.49 | 1.08 | 0.77 |
| Ours-tiny | 0.98 | 0.85 | 0.48 | 0.47 | 1.01 | 0.76 |
| Ours(TrajCLIP) | 0.84 | 0.79 | 0.45 | 0.41 | 0.90 | 0.68 |

| FDE | A2B | B2C | C2D | D2E | E2A | AVG |
|---|---|---|---|---|---|---|
| tra2tra | 3.29 | 2.50 | 1.78 | 0.92 | 2.18 | 2.13 |
| T-GNN | 2.18 | 1.82 | 1.30 | 0.65 | 1.86 | 1.56 |
| K0 | 1.83 | 1.67 | 0.92 | 1.01 | 2.14 | 1.51 |
| Ours-tiny | 1.92 | 1.66 | 0.94 | 0.94 | 1.99 | 1.49 |
| Ours(TrajCLIP) | 1.70 | 1.61 | 0.91 | 0.73 | 1.87 | 1.36 |

Table 2: **Left** is [ETH-UCY → DeathCircle_0 in SDD] Comparison of ADE reduction of each method for updating models based on online observed data. **Right** is prediction performance in scene-to-scene transfer experiment setting on ETH-UCY using ADE and FDE. A, B, C, D, E denote ETH, Hotel, Univ, Zara1, Zara2. A2B experiment denotes a training model with a training set from A as well as the validation set from B, and reports test performance on a test set from B.

**Few-shot Transfer Task**. In this experiment, different from the scene-to-scene transfer, we train the model with the entire ETH-UCY dataset and four trajectories in the object scene in SNU dataset and evaluate with the other trajectories from the object scene. This experimental setting is called 4-shot transfer learning. We compare our TrajCLIP and TrajCLIP-tiny with several models: MemoNet [26], performs trajectory prediction with a memory-driven mechanism; Social-STGCNN [16], which claims data efficiency and performs well with 20% data in a dataset. T-GNN [29] and K0 [10] are models designed especially for the transfer learning task. The experiment results on ADE/FDE are shown in right part of Table 3. Only K0 outperforms our TrajCLIP-tiny 2.5%, but TrajCLIP-tiny performance remains competitive with other methods. With the ability to model more complex distributions, the TrajCLIP method outperforms all other methods on each SNU scene 8.7% to 40.6%.

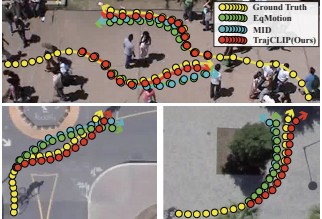

|  | Oneway | Stagger | Stroll | AVG |
|---|---|---|---|---|
| MemoNet | 0.31/0.62 | 0.34/0.65 | 0.32/0.59 | 0.32/0.62 |
| Social-STGCNN | 0.34/0.69 | 0.37/0.72 | 0.40/0.79 | 0.37/0.73 |
| T-GNN | 0.28/0.51 | 0.32/0.58 | 0.34/0.57 | 0.31/0.55 |
| K0 | 0.27/0.48 | 0.33/0.61 | 0.31/0.53 | 0.30/0.54 |
| Ours-tiny | 0.29/0.53 | 0.33/0.59 | 0.32/0.54 | 0.31/0.55 |
| Ours(TrajCLIP) | **0.26/0.47** | **0.31/0.56** | **0.28/0.44** | **0.28/0.49** |

Table 3: **Left** is qualitative results of prediction results on UNIV scene (top) of ETH-UCY dataset, little_3 scene (left) and coupa scene (right) of SDD dataset. **Right** is ADE/FDE comparison of 4-shot transfer learning performance on each SNU scene.

**Online Adaption Task**. We evaluate the model performance of the online adaptation task. All methods are trained on the ETH-UCY dataset initially, with a batch size of 1 to simulate an online situation where data are observed sequentially. Then, all models are online adapted and tested. Our model adopts the online adaptation in Sec. 4.2 and compares it with the meta-learning-based K0 with/without additional online fine-tuning, as well as other methods that have better performance in online tasks. As shown in the left figure of Table 2, we take DeathCircle_0 in SDD as a new scenario as an example to give our comparison results on the online adaptation task. We select DeathCircle_0 in SDD as the target scene and compare the online adaptation effects of each model in the left figure of Table 2. The term "base" in the figure denotes the results of offline training for each model on DeathCircle_0. Using this prediction error as a benchmark, we calculate the ADE reduction rate of each model during the online adaptation process. Our model performance decreases the least at first and is improving fastest as the training step increases. After 100 steps, our model performs 20% better than the base model, which proves it could adapt to predict trajectories in new scenes quickly.

|  | SAIF | CLIP | Generation framework | ETH-UCY | SDD |
|---|---|---|---|---|---|
| (a) | ✗ | ✓ | IGN | 0.20/0.36 | 6.34/12.08 |
| (b) | ✓ | ✗ | IGN | 0.24/0.50 | 7.85/14.66 |
| (c) | ✓ | ✗ | CVAE | 0.24/0.47 | 7.91/15.03 |
| (d) | ✓ | ✓ | CVAE | 0.23/0.44 | 7.86/14.62 |
| (e) | ✓ | ✗ | Diffusion | 0.22/0.39 | 7.39/13.95 |
| (f) | ✓ | ✓ | Diffusion | 0.21/0.38 | 7.25/13.84 |
| Ours | ✓ | ✓ | IGN | 0.18/0.33 | 6.29/11.79 |

| method | SP | API | PTG | ETH-UCY | SDD |
|---|---|---|---|---|---|
| EqMotion | ✗ | ✗ | ✗ | 0.21/0.35 | 7.52/12.88 |
|  | ✓ | ✓ | ✓ | 0.20/0.35 | 7.34/12.67 |
| TUTR | ✗ | ✗ | ✗ | 0.21/0.36 | 7.76/12.69 |
|  | ✓ | ✓ | ✓ | 0.20/0.34 | 7.41/12.47 |
| TrajCLIP | ✓ | ✗ | ✗ | 0.20/0.35 | 6.89/13.06 |
|  | ✗ | ✓ | ✗ | 0.19/0.34 | 6.51/12.32 |
|  | ✗ | ✗ | ✓ | 0.19/0.35 | 6.88/12.95 |
|  | ✗ | ✓ | ✓ | 0.18/0.34 | 6.46/12.06 |
| Ours(TrajCLIP) | ✓ | ✓ | ✓ | 0.18/0.33 | 6.29/11.79 |

Table 4: Ablation study on each key module of our TrajCLIP model (left). Ablation study on each data augmentation (right). SP denotes Small Perturbations, API denotes Adjacent Point Interpolation, PTG denotes Plausible Trajectories Generation.

**Ablation Studies.** In left part of Table 4, we present the performance contributions of our proposed key modules: the SAIF, the CLIP pairing learning, and the manifold predictor (IGN generation module). The results show that combining the CLIP pairing method and the IGN module significantly enhances prediction performance, surpassing all other generation methods. This is because the CVAE framework involves an encoder obtaining a feature and then randomly sampling to get the mean and variance, which are used as inputs for the decoder. The diffusion method starts with a random initialization of an input and then gradually denoises. This randomness makes alignment work less effective (that is, even if aligned, the presence of random sampling still makes the generated results uncontrollable). The comparative experiments (c)(d) and (e)(f) illustrate that the two random methods of CVAE and diffusion methods make the improvement effect of alignment work not significant. On the ETH_UCY dataset, the addition of CLIP only improved ADE/FDE by 0.01/0.03 and 0.01/0.01. However, the network structure of IGN is an MLP, which is an affine transformation and does not have this issue; hence, as in our method, alignment is significantly effective on IGN. When CLIP is removed, as shown in experiments (b)(c)(e), since the two feature spaces cannot be connected through IGN without alignment, the performance drops by 0.06/0.17 compared to when CLIP is added.

We also conduct experiments to test different data augmentation methods and find that our method's improved performance is due to TrajCLIP's strong modeling ability to represent complex trajectory distributions, which shown in right part of Table 4. The results of applying the data augmentation methods are shown in the rightmost column of Table. When we train the latest state-of-the-art methods EqMotion and TUTR with our proposed data augmentation methods, we observe minimal changes in their performance, whereas our model's performance improved significantly. Additionally, experiments on each of the three core data augmentation methods demonstrate their effectiveness. Among these methods, Adjacent Point Interpolation (API) is proved to be the most effective in enhancing the model's prediction accuracy.

| | Model Size (MB) | Computational Complexity (GFlops) | Infer Speed (s) |
|---|---|---|---|
| trajectron++ | 0.53 | 2.48 | 0.0223 |
| AgentFormer | 6.78 | 12.79 | 0.1874 |
| MID | 2.42 | 9.06 | 0.8802 |
| Y-net | 203.23 | 35.70 | 1.0716 |
| TUTR | 0.46 | 3.51 | 0.0577 |
| Ours-tiny | 3.45 | 5.26 | 0.0615 |
| Ours(TrajCLIP) | 14.94 | 18.96 | 0.2108 |

Table 5: Comparison of our method with other methods in terms of model size, computational complexity, and infer speed. Infer speed refers to the time required to predict the next 12 frames using 8 frames.

**Infer Speed and Computation Consumption.** As the table 5 illustrates, we have contrasted our approach with alternative techniques in terms of model size, computational complexity, and inference speed. Our medium-sized model meets the requirements for a real-time prediction task in terms of both inference speed and model size, as it can predict trajectories in 0.0615 seconds. Additionally, our lightweight model is only 3.45MB in size, and its computational complexity is relatively low compared to its model size, making it deployable on most hardware platforms.

**Qualitative Results.** The left figure of Table 3 shows the visualization results of trajectory prediction for complex history trajectory using TrajCLIP and other state-of-the-art methods EqMotion and MID on UNIV scene of ETH-UCY dataset. Two trajectories in the figure include interference of surrounding pedestrians and scene environment and several turnings. We see that our TrajCLIP could predict these unusual turns, and the other two methods only predict a common smooth trajectory pattern, which shows a better ability to model complex trajectories.

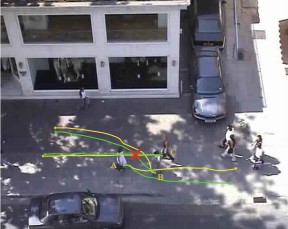 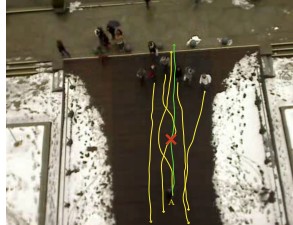 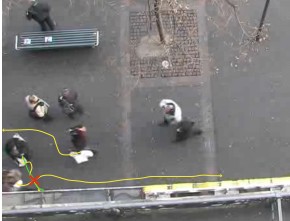

Figure 5: Visualization of failure cases on the ETH/UCY dataset. Given the observed historical trajectories (yellow), our method predicts future trajectories (green). In complex interaction scenarios, our method occasionally exhibits failures such as deviating from the intended endpoint and colliding (red) with other agents.

**Limitation Analysis.** as illustrated in Figure 5, our method falls short in ensuring collision avoidance in complex, high-density environments, necessitating further research.

## 5 Conclusion

In our work, we introduce TrajCLIP, a method for predicting pedestrian trajectories. We use contrastive learning and an idempotent generative network to model a complex trajectory feature space. TrajCLIP offers five key contributions: the ability to model intricate trajectory distributions, CLIP pairing learning to combine the feature spaces of historical and future trajectories, a trajectory feature encoder based on the fusion of time domain and frequency domain, an idempotent generative network for mapping the same feature space, and an adjacent point interpolation algorithm to enhance the model's expression ability. We evaluate TrajCLIP on four prediction tasks, including standard trajectory prediction, scene-to-scene transfer, few-shot transfer, and online adaption, using three datasets. Our method achieves state-of-the-art prediction performance.

## 6 Acknowledgement

This work was supported by the National Natural Science Foundation of China under Grant 62002345.

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
