# OpenReview forum: "TrajCLIP: Pedestrian trajectory prediction method using contrastive learning and idempotent networks"
_NeurIPS.cc/2024/Conference — NeurIPS 2024 poster_

### Official Review · Reviewer_iNsW · 2024-07-10

**Soundness:** 3
**Presentation:** 3
**Contribution:** 3
**Rating:** 6
**Confidence:** 4

**Summary:**

The paper proposes a trajectory prediction approach for multi-agent configurations. The historical and future trajectories are encoded in terms of spatio-temporal interaction features (STIF) using Agentformer [28] and scene-agent interaction features (SAIF) using a transformer based on Fourier transform. The historical and future trajectory encoders share the same architecture but have different parameters. The encoded features are then summed and passed to a generative model [21] to represent  the trajectory space. Finally, a transformer predicts the next position. The encoded features for historical and future trajectories (STIF and SAIF) are trained using CLIP [16] to force the historical and future encoded features to share the same space and help achieve temporal continuity for the trajectory prediction task. The approach is evaluated on the ETH-UCY, SDD and SNU datasets and shows promising results compared to related methods.

**Strengths:**

- The paper is well written and easy to follow. In addition, the approach is well presented and the evaluation is clearly described.
- The proposed approach achieves promising results on several standard benchmarks. The results are convincing.
- The paper provides a wealth of ablation studies.
- The proposed combination of coding and decoding is interesting and makes a solid contribution. In particular, the adaptation of CLIP from image-text pairs to the historical-future trajectory feature space is novel.

**Weaknesses:**

- The paper makes various solid and small contributions, but it lacks a novel idea. This is a limitation, but it's not a major point.
- The related work would benefit from a more direct comparison of the proposed method with the related works.
- The motivation for relying on an idempotent generative network over other generative models, e.g. GANs, is not well explained.
- According to the literature, the article "Human Trajectory Prediction via Neural Social Physics" by Yue et. al. (2022) achieves average ADE and FDE of 0.17 and 0.24 respectively on ETH, while 6.52 ADE and 10.61 FDE are achieved on SDD. The comparisons should include all state-of-the-art methods.

**Questions:**

- For the Online Adaptation task, prediction times are another crucial factor to consider alongside performance metrics. Is there any reason not to report them?
- Could the authors provide details of the hyper-parameters used during training? Implementation details are missing
- There are significant discrepancies between the results you report in Table 1 for Traj++ and the results reported in other publications, such as Traj++ in "Human Trajectory Prediction via Neural Social Physics". Could the authors provide more details on the experimental setup?

**Limitations:**

The paper does not dedicate space to discuss the limitations. This is a clear missing point.

---

> ### Author Rebuttal · Authors · 2024-08-07
>
> > Q1:The motivation for relying on an idempotent generative network over other generative models, e.g. GANs, is not well explained.
>
> Further explanation on the starting point for choosing the idempotent generation framework. Our motivation is that, for trajectory prediction task, the trajectory features of an agent should be consistently aligned in the feature space of both historical and future trajectories. However, GANs and CVAEs (SocialGAN) both have separate encoding-decoding structures, and diffusion inference samples the initial space for generation, which leads to a certain degree of inconsistency in the feature space. The idempotent generation framework is an affine transformation in the feature space, which can effectively bridge the gap in the feature space caused by the generation framework. In addition, the idempotent neural framework has the characteristic of global projection, which can more effectively improve generalization ability, allowing our method to perform well on different tasks without fine-tuning. Our extensive quantitative and qualitative experiments have fully demonstrated this point. Please refer to Reviewer  AYPL Q3 for more experiential analysis.
>
> > Q2: article NSP-SFM[1] performs better. The comparisons should include all state-of-the-art methods.
>
> Further explanation on the comparison with the SOTA methods, we will compare and analyze with the methods mentioned by the reviewers in the final version. (1) We acknowledge that the model in the article NSP-SFM[1] has better predictive quantitative performance than TrajCLIP. However, the NSP-SFM method first relies on a CNN-based network to extract goal information from the map, which is not feasible for many complex scene prediction tasks as the goal information is not available. Secondly, the method's estimation capability for multimodal trajectories is not achieved by a generative model and does not consider the diversity of pedestrian intentions during the prediction process; moreover, the network has a large number of parameters, with its model parameter count being more than 50 times that of other trajectory prediction networks. In contrast, TrajCLIP's predictions do not need to rely on goal information and can be predicted independently, without being limited by the prediction anchor points, and with a smaller number of parameters. We are more focused on comparing with similar methods to demonstrate the model's performance and capabilities in the context of the predictive task's focus points.
>
> We compared four recent trajectory prediction state-of-the-art method from CVPR 2024 in Table 4.  our method achieves the best ADE/FDE results on the ETH-UCY dataset.
>
> > Q3:For the Online Adaptation task, prediction times are another crucial factor to consider alongside performance metrics. Is there any reason not to report them?
>
> We have included implementation details in Table 1 in attached PDF. Our tiny model meets the requirements for a real-time prediction, as it can predict trajectories in 0.0615 seconds. Additionally, our lightweight model is only 3.45MB in size, and its computational complexity is relatively low compared to its model size, making it deployable on most hardware platforms.
>
> > Q4:Could the authors provide details of the hyper-parameters used during training? Implementation details are missing.
>
> For training process, our batch size was set to 64, epochs to 100, and the learning rate was 0.01, which was half-formed every 25 epochs. We used the Adam optimizer. Our model was trained on an RTX 3090, with the encoder and contrastive learning pre-training requiring approximately 8 GPU hours, and the full framework training taking about 9.6 GPU hours.
>
> > Q5: There are significant discrepancies between the results you report in Table 1 for Trajectron++ and the results reported in other publications, such as Trajectron++ in "Human Trajectory Prediction via Neural Social Physics". Could the authors provide more details on the experimental setup?
>
> Explanation of the reproduction details for Trajectron++. When reproducing Trajectron++[3], there was a data leakage issue in the original released code implementation, leading to better metrics. The original author also acknowledged this on GitHub issue #26. "The function used for differentiating is **derivative of** and calls **np.gradient**. np.gradients approximates in different ways for boundary points and non-boundary points. For an example of calculating velocities, np.gradients will give $v _x[t] = (x[t+1]-x[t-1])/(2*dt)$ at time t." This method inadvertently includes future trajectory information in the last frame of historical data, which significantly affects the model's predictive performance. The author of Trajectron++ suggests using the correct differentiation method, such as **np.ediff1d**, to preprocess velocity and acceleration inputs to avoid data leakage caused by using information beyond the last observed time step. (Due to conference policy restrictions, we cannot provide a link.) After resolving this data issue, we reproduced the experiment using the method suggested by the author, which resulted in differences in the cited values when compared with other papers. In addition, the recent CVPR 2024 paper: SingularTrajectory [2] and many other works also adopt the same practice when citing the Trajectron++ work, and the metrics are consistent.
>
> [1] Yue, Jiangbei, Dinesh Manocha, and He Wang. "Human trajectory prediction via neural social physics." European conference on computer vision. Cham: Springer Nature Switzerland, 2022.
>
> [2]  Bae, Inhwan, et al. "SingularTrajectory: Universal Trajectory Predictor Using Diffusion Model." CVPR. 2024.
>
> [3] Salzmann, Tim, et al. "Trajectron++: Dynamically-feasible trajectory forecasting with heterogeneous data." Computer Vision–ECCV 2020: 16th European Conference, Glasgow, UK, August 23–28, 2020, Proceedings, Part XVIII 16. Springer International Publishing, 2020.

---

> > ### Comment · Reviewer_iNsW · 2024-08-12
> >
> > The rebuttal does a very good job of addressing my concerns and most of the other reviewers' points. For this reason, I am moving to a positive rating (updated scores).

---

### Official Review · Reviewer_NFeW · 2024-07-10

**Soundness:** 2
**Presentation:** 3
**Contribution:** 3
**Rating:** 5
**Confidence:** 4

**Summary:**

This paper proposes to utilize contrastive learning for pedestrian trajectory prediction. A STIF encoder is used to extract spatial-temporal features and is trained with data augmentation. A SAIF utilizes the Fast Fourier Transform to extract the interaction information among the agents. The authors incorporate the idempotent loss for training. Experiments on three datasets show the efficacy of the proposed method.

**Strengths:**

1. The idea of using contrastive learning and idempotent loss is interesting.
2. The performance is good. The proposed method achieves significantly better results than SOTA on SDD and others.

**Weaknesses:**

1. I have some doubts about the intuition of using CLIP between history and future trajectories. See questions below.
2. There is no speed or computational cost comparison. Since the method involves multiple stages, a comparison of training/inference costs is needed.
3. The proposed method does not consider visual scene information.
4. There is no error analysis. When does the method fail and why?

Minor comments:
Section 2.2 Generalization Framework -> Generative Framework. They are different things.
The text in Figure 2 is too small to read.
Line 263 “We conducted comparative ”, Line 270 “we compare our model”. Better to use a consistent present tense when describing your method.

**Questions:**

1. I have some doubts about the intuition of using CLIP learning between the historical feature and the future trajectory feature. The authors propose to align historical feature space with future feature space (Line 186). Does this lead to identical trajectory prediction, i.e., copying the historical trajectory as the future trajectory? Different from aligning images and text that have the same semantic meaning, the historical trajectory and future trajectory of the same agent may be different due to the difference in the spatial location (hence the surroundings are different).
2. Since the method involves multiple stages, how does the model compare to the baselines in terms of inference speed and training cost?
3. There is no error analysis. When does the method fail and why?

**Limitations:**

The authors did not discuss the limitations of the method and failure cases.

---

> ### Author Rebuttal · Authors · 2024-08-07
>
> > Q1:I have some doubts about the intuition of using CLIP between history and future trajectories.
>
> Regarding further clarification on the unified encoding of historical and future trajectories using CLIP. In our method, the trajectory encoders are designed to capture motion characteristics rather than static spatial data, enabling synchronization of feature spaces between the future and historical trajectory encoders. Considering that inputs for pedestrian trajectory prediction consist of sequential relative coordinates, the encoders focus on the agent's trajectory, surrounding dynamics, and their interactions. The decoder then infers future locations from these features. For datasets like ETH\_UCY, where 20 frames equal 5 seconds, spatial consistency is maintained by analyzing the same two frames. The use of STIF and SAIF is for modeling trajectory and interaction features, respectively, aligning them in the feature space to prevent redundancy in trajectory predictions that could arise from spatial alignment.
>
> > Q2: There is no speed or computational cost comparison. Since the method involves multiple stages, a comparison of training/inference costs is needed.
>
> For training process, our batch size was set to 64, epochs to 100, and the learning rate was 0.01, which was half-formed every 25 epochs. We used the Adam optimizer. Our model was trained on an RTX 3090, with the encoder and contrastive learning pre-training requiring approximately 8 GPU hours, and the full framework training taking about 9.6 GPU hours.
>
> For inference process, as shown in Table 1, our medium-sized model meets the requirements for a real-time prediction task in terms of both inference speed and model size, as it can predict trajectories in 0.0615 seconds. Additionally, our lightweight model is only 3.45MB in size, and its computational complexity is relatively low compared to its model size, making it deployable on most hardware platforms.
>
> > Q3:The proposed method does not consider visual scene information.
>
> Thank you for the new perspective provided by the reviewer. Our focus lies in how to align the trajectory features and interaction features of agents in the historical and future spaces within trajectory prediction, and we are concerned with whether this method can be generalized to various subtasks at a low cost. Due to the high real-time requirements of autonomous driving, we chose not to introduce image information. Admittedly, using the features of visual information from map scenes to improve the performance of trajectory prediction is indeed another angle worth paying attention to. We will consider how to efficiently integrate visual information to further enhance the predictive effect of the model in our subsequent work.
>
> > Q4: Minor comments: Section 2.2 Generalization Framework -> Generative Framework. They are different things. The text in Figure 2 is too small to read. Line 263 “We conducted comparative ”, Line 270 “we compare our model”. Better to use a consistent present tense when describing your method.
>
> We will address these issues (e.g., typos, grammar mistakes) in the camera-ready version.
>
> > Q5:There is no error analysis. When does the method fail and why?
>
> We express our gratitude for the feedback. Due to spatial limitations, a summary of the limitations of our methodology will be presented in the camera-ready version. Our approach primarily focusses on the modelling of trajectory prediction tasks, in line with common practices that omit scene imagery as input. Therefore, empirical validation of the model's performance in practical scenarios is essential. As illustrated in Figure 1 in attached PDF, our method falls short in ensuring collision avoidance in complex, high-density environments, necessitating further research.

---

### Official Review · Reviewer_AYPL · 2024-07-10

**Soundness:** 3
**Presentation:** 3
**Contribution:** 3
**Rating:** 6
**Confidence:** 3

**Summary:**

This paper utilizes idempotent generative network to perform multiple tasks in pedestrian trajectory prediction and achieves state-of-the-art performance in those tasks, showing its great representation and generalization ability.

The proposed model has the following main components:
1. Spatio-Temporal Interaction Feature (STIF) and Scene-Agent Interaction Feature (SAIF) based encoder
2. CLIP for past and future feature alignment
3. IGN for mapping from past to future in aligned feature space.

The authors also provide a theoretical analysis of IGN, CLIP and data augmentation, but some issues need to be addressed.

**Strengths:**

1. First-time usage of IGN in pedestrian trajectory prediction task, which is novel.
2. The paper is well-written and easy to follow.
3. Experiment on various kinds of pedestrian trajectory prediction tasks is sufficient and a universal model for those tasks is the main tendency.

**Weaknesses:**

1. This is not the first work to use constructive learning in pedestrian trajectory prediction, since it was used in long-tailed pedestrian trajectory prediction for forming better latent representations [1, 2]. The author should cite them and clarify the difference.
2. Missing citation and comparison for SingularTrajectory[3], which is also a universal model for pedestrian trajectory prediction. It defines a singular space that can well represent trajectory features for multiple tasks without alignment between past and future.
3. Some questions need to be addresses. See questions.



[1] Wang, Yuning, Pu Zhang, Lei Bai, and Jianru Xue. "Fend: A future enhanced distribution-aware contrastive learning framework for long-tail trajectory prediction." In Proceedings of the IEEE/CVF Conference on Computer Vision and Pattern Recognition, pp. 1400-1409. 2023.

[2] Makansi, Osama, Özgün Cicek, Yassine Marrakchi, and Thomas Brox. "On exposing the challenging long tail in future prediction of traffic actors." In Proceedings of the IEEE/CVF International Conference on Computer Vision, pp. 13147-13157. 2021.

[3] Bae, Inhwan, Young-Jae Park, and Hae-Gon Jeon. "SingularTrajectory: Universal Trajectory Predictor Using Diffusion Model." In Proceedings of the IEEE/CVF Conference on Computer Vision and Pattern Recognition, pp. 17890-17901. 2024.

**Questions:**

1. Alignment is quite useful for IGN but not as much useful as IGN when applying to CVAE and diffusion according to ablative experiments. Why was that? This issue needs more analysis and clarification.
From the ablative experiments, when CLIP is not used, IGN is worse than CVAE and diffusion (from b, d, f in Tab.4). Meanwhile, CLIP cannot provide pronounced improvement as IGN when applied on CVAE and diffusion. The author just says combining them achieves the best results but motivation of combining IGN and CLIP is not very clear. Is it due to some intrinsic characters of IGN that requires feature alignment while other models do not drop as much as IGN when removing it? More detailed analyses of the phenomena is needed.

2. Is the affine transformation mentioned in Fig.1 same as the trainable matrix W in CLIP?

**Limitations:**

Not seen in main content.

---

> ### Author Rebuttal · Authors · 2024-08-07
>
> > Q1: This is not the first work to use constructive learning.
>
> We thanks reviewers for providing a new perspective. The mentioned two works, long-tail analysis[1] and FEND[2], both utilize contrastive learning, but our work differs in the problems it addresses with contrastive learning. These two mostly use contrastive learning to capture features from historical trajectories. They aim to get better representations of historical trajectory encodings in latent space, which will improve their ability to predict trajectories for the long-tail part of the dataset.}
>
> For example, long-tail analysis uses a Kalman filter to directly predict output trajectory errors as a basis, distinguishing difficult and easy samples. It then conducts contrastive learning on difficult (long-tail part) historical trajectories to enhance the feature representation capabilities of the long-tail part; FEND uses VAE to extract features from the entire trajectory and performs offline clustering on the features to obtain labels; during training, it uses offline labels for contrastive learning on historical trajectories to guide the feature generation of historical trajectories (making features extracted from more similar historical trajectories closer), enhancing the prediction effect of the long-tail part.
>
> However, our work, not only uses contrastive learning for better encoding of historical trajectories but also acts on future trajectories. While ensuring historical and future motion consistency, it achieves a more unified latent space representation of future and historical trajectories, thereby enhancing the model's predictive generalization capability.
>
> > Q2: Missing citation and comparison for SingularTrajectory.
>
> SingularTrajectory[3] introduces a diffusion model for multimodal pedestrian trajectory prediction, utilizing singular space rather than trajectory coordinates to convey motion information and capture the dynamics of trajectories. It also employs an adaptive anchor-based strategy for dynamically adjusting trajectory anchors. We acknowledge the merits of this technique, but our approach surpasses this work in terms of performance and adaptability. The summary is as follows:
>
> (1) In the performance comparison on the ETH-UCY dataset as shown in Table 3, our model achieves a 14\% lower Average Displacement Error (ADE) than the SingularTrajectory model (0.18 for our model versus 0.21 for SingularTrajectory), with the difference in Final Displacement Error (FDE) being negligible.
>
> (2) For transfer learning and few-shot learning tasks, given the differences in task settings and data processing between our method and SingularTrajectory, we conducted experiments following the experimental framework described in their paper. The results of these experiments are presented in Tables. 2 and 3. Observations indicate that our model attains comparable ADE performance to SingularTrajectory in transfer learning tasks and slightly outperforms in FDE. In few-shot learning tasks, our model demonstrates superior performance in both ADE and FDE.
>
> (3) Regarding the task of online adaptability, the architecture of SingularTrajectory, which incorporates complex diffusion models and scene semantic segmentation techniques, results in slower inference speed, making online learning impractical. In contrast, our model consistently outperforms in this task.
>
> > Q3:More detailed analyses for ablation experimental phenomena about CLIP and generative framework (IGN, CVAE, diffusion).
>
> We further explain the analysis of the ablation study. The CVAE framework involves an encoder obtaining a feature and then randomly sampling to get the mean and variance, which are used as inputs for the decoder. The diffusion method starts with a random initialization of an input and then gradually denoises. This randomness makes alignment work less effective (that is, even if aligned, the presence of random sampling still makes the generated results uncontrollable). As shown in Table 4 in main content, the ablation experiments of CLIP and the generation framework were conducted. The comparative experiments (c)(d) and (e)(f) illustrate that the two random methods of CVAE and diffusion methods make the improvement effect of alignment work not significant. On the ETH\_UCY dataset, the addition of CLIP only improved ADE/FDE by 0.01/0.03 and 0.01/0.01. However, the network structure of IGN is an MLP, which is an affine transformation and does not have this issue; hence, as in our method, alignment is significantly effective on IGN. When CLIP is removed, as shown in experiments (b)(c)(e), since the two feature spaces cannot be connected through IGN without alignment, the performance drops by 0.06/0.17 compared to when CLIP is added.
>
> > Q4:Is the affine transformation mentioned in Fig.1 same as the trainable matrix W in CLIP?
>
> In our approach, CLIP is only utilized to facilitate the alignment of feature spaces during the pre-training phase. The affine transformation of the idempotent generative network operates independently of the parameters of CLIP. However, using CLIP's trainable matrix W as an initialization for the network's parameters could enhance the generative framework. Since CLIP's trainable matrix W can project the feature space into a unified fusion space, it is possible to employ this matrix W as an initial parameter within the idempotent generative framework and subsequently further train the idempotent generative network based on it. This is a perspective worth exploring, and we will conduct further research on this attempt in the subsequent studies.
>
> [1] Makansi, Osama, et al. "On exposing the challenging long tail in future prediction of traffic actors." ICCV. 2021.
>
> [2] Wang, Yuning, et al. "Fend: A future enhanced distribution-aware contrastive learning framework for long-tail trajectory prediction." CVPR. 2023.
>
> [3] Bae, Inhwan, et al. "SingularTrajectory: Universal Trajectory Predictor Using Diffusion Model." CVPR. 2024.

---

> > ### Comment · Reviewer_AYPL · 2024-08-14
> >
> > Thank you for the author's response. The reply addressed most of my concerns, and I am inclined to maintain my original score.

---

### Official Review · Reviewer_Xn2A · 2024-07-14

**Soundness:** 3
**Presentation:** 3
**Contribution:** 3
**Rating:** 5
**Confidence:** 3

**Summary:**

This paper presents TrajCLIP, a novel method for pedestrian trajectory prediction that utilizes contrastive learning and idempotent neural networks. The authors propose an interesting approach to address some limitations of existing methods, particularly in terms of generalization and modeling complex trajectory distributions.

**Strengths:**

1 The paper introduces an innovative idea of using contrastive learning to align the feature spaces of historical and future trajectories. This approach has the potential to improve the model's ability to generalize across different scenarios.

2 The use of idempotent neural networks for global feature mapping is a creative solution to prevent overfitting to specific dataset distributions.

3 The combination of time-domain and frequency-domain features in the trajectory encoder is an interesting approach that could capture more comprehensive trajectory information.

**Weaknesses:**

1 The paper lacks a discussion on the computational complexity and runtime performance of the proposed method compared to existing approaches.

2 More analysis of the limitations of the proposed method and potential failure cases would strengthen the paper.

**Questions:**

1 Provide more detailed explanations of the idempotent neural network implementation and training process.

2 Add a section discussing the computational requirements and runtime performance of TrajCLIP.

3 Include a discussion on the limitations of the proposed method and potential scenarios where it might not perform well.

**Limitations:**

The limitations discussed in the Experiments seem too easy and do not fully reflect the limitations of the work.

---

> ### Author Rebuttal · Authors · 2024-08-07
>
> > Q1: Add a section discussing the computational requirements and runtime performance of TrajCLIP.
>
> We appreciate you bringing up the model performance experiment. As the table illustrates, we have contrasted our approach with alternative techniques in terms of model size, computational complexity, and inference speed. Our medium-sized model meets the requirements for a real-time prediction task in terms of both inference speed and model size, as it can predict trajectories in 0.0615 seconds. Additionally, our lightweight model is only 3.45MB in size, and its computational complexity is relatively low compared to its model size, making it deployable on most hardware platforms.
>
> |                | Model Size (MB) | Computational Complexity (GFlops) | Infer Speed (s) |
> |----------------|:---------------:|:---------------------------------:|:---------------:|
> | trajectron++   |       0.53      |               2.48               |      0.0223     |
> | AgentFormer    |       6.78      |              12.79               |      0.1874     |
> | MID            |       2.42      |               9.06               |      0.8802     |
> | Y-net          |      203.23     |              35.70               |      1.0716     |
> | TUTR           |       0.46      |               3.51               |      0.0577     |
> | Ours-tiny      |       3.45      |               5.26               |      0.0615     |
> | Ours(TrajCLIP) |      14.94      |              18.96               |      0.2108     |
>
> *Comparison of our method with other existing methods in terms of model size, computational complexity, and inference speed. Inference speed refers to the time required to input an 8-frame trajectory and predict the next 12 frames.*
>
> > Q2:More analysis of the limitations of the proposed method and potential failure cases would strengthen the paper.
>
> We express our gratitude for the feedback. Due to spatial limitations, a summary of the limitations of our methodology will be presented in the camera-ready version. Our approach primarily focusses on the modelling of trajectory prediction tasks, in line with common practices that omit scene imagery as input. Therefore, empirical validation of the model's performance in practical scenarios is essential. As illustrated in Figure 1 in attached PDF, our method falls short in ensuring collision avoidance in complex, high-density environments, necessitating further research.
>
> > Q3: Provide more detailed explanations of the idempotent neural network implementation and training process.
>
> We will provide a detailed summary of the model's experimental details and training process in the camera-ready version and will also release the source code. The experimental details of the idempotent neural network can be summarised as follows: First, we froze the pre-trained historical trajectory encoder and trained the manifold predictor, that is, the idempotent neural network, as well as the manifold decoder. The manifold predictor was trained using reconstruction loss, idempotent loss, and tightness loss, while the manifold decoder was trained using L2 loss. Training the idempotent and tightening losses requires gradient clipping. For clarity, we provide the training code as follows. Moreover, an RTX 3090 was used to train our model. Our training parameters included a 64-batch size, 100 epochs, and a 0.01 learning rate that was half-formed every 25 epochs. The Adam optimizer was employed.
>
> ```python
> def ign_train(f, f_copy, Z_H):
>     # f, f_copy : MLP_θ
>     predict_Z_F = f(Z_H)
>
>     f_z = Z_F.detach()
>     ff_z = f(f_z)
>     f_fz = f_copy(fz)
>
>     # calculate losses
>     loss_rec = (predict_Z_F - Z_F).pow(2)
>     loss_idem = (f_fz - fz).pow(2)
>     loss_tight = -(ff_z - f_z).pow(2)
>
>     # optimize for losses
>     loss = loss_rec + loss_idem + loss_tight * 0.1
>     opt.zero_grad()
>     loss.backward()
>     opt.step()
> ```
>
> [1] Shocher, Assaf, et al. "Idempotent Generative Network." The Twelfth International Conference on Learning Representations.

---

> ### Comment · Reviewer_Xn2A · 2024-08-14
>
> The author's response has resolved some of my questions, but I still have questions about the Computational Complexity and Inference Speed of the paper. Compared to the previous state-of-the-art papers, this paper has significantly increased the number of parameters, but the performance improvement is limited. So I will maintain my "Borderline accept" score for this paper.

---

### Author Rebuttal · Authors · 2024-08-07

Thank you to all reviewers for your valuable comments and recognition of the novelty of our work. We have provided further elaboration and clarification in response to the reviewers' feedback, along with additional experiments to supplement the explanation. We will address each reviewer's comments and queries individually to assist in understanding this work. And we will release the source code and address all the reviewers' issues in the camera-ready version.

We have included more experiment results in the attached file, to facilitate the reviewers understanding of our responses.

Here, we list the additional experiments according to each reviewer's comments.

**[Xn2A-Q2, AYPL, NFeW-Q5, iNsW]** limitation and potential failure discussion:  we have conducted supplemental experiments and discussed the limitation and failure cases of our method as shown in Figure 1.

**[Xn2A-Q1, NFeW-Q2]** model computational complexity discussion: We have conducted additional experiments and discussed the comparison of model size, computational complexity and inference speed between our TrajCLIP and other recent popular pedestrian trajectory prediction models, as shown in Table 1.

**[AYPL-Q2]** comparison for SingularTrajectory[1]: We compare our work with SingularTrajectory from transfer learning and few-shot learning, as shown  in Table 2 and Table 3.

**[iNsW-Q3]** comparisons with all state-of-the-art methods: we compare our work with multiple SOTA methods[1-4], as shown in Table 4.

|                | Model Size (MB) | Computational Complexity (GFlops) | Infer Speed (s) |
|:--------------:|:---------------:|:---------------------------------:|:---------------:|
| trajectron++   |       0.53      |               2.48               |      0.0223     |
| AgentFormer    |       6.78      |              12.79               |      0.1874     |
| MID            |       2.42      |               9.06               |      0.8802     |
| Y-net          |      203.23     |              35.70               |      1.0716     |
| TUTR           |       0.46      |               3.51               |      0.0577     |
| Ours-tiny      |       3.45      |               5.26               |      0.0615     |
| Ours(TrajCLIP) |      14.94      |              18.96               |      0.2108     |

*Tab1. Comparison of our method with other existing methods in terms of model size, computational complexity, and inference speed. Inference speed refers to the time required to input an 8-frame trajectory and predict the next 12 frames.*

| ADE                |  A2B  |  A2C  |  A2D  |  A2E  |  AVG  |
|:------------------:|:-----:|:-----:|:-----:|:-----:|:-----:|
| SingularTrajectory |  0.29 |  0.59 |  0.51 |  0.42 |  0.45 |
| Ours(TrajCLIP)     |  0.30 |  0.59 |  0.49 |  0.43 |  0.45 |

| FDE                |  A2B  |  A2C  |  A2D  |  A2E  |  AVG  |
|:------------------:|:-----:|:-----:|:-----:|:-----:|:-----:|
| SingularTrajectory |  0.57 |  1.19 |  1.08 |  0.81 |  0.91 |
| Ours(TrajCLIP)     |  0.59 |  1.20 |  0.99 |  0.83 |  0.90 |

*Tab 2. Comparison of ADE/FDE for transfer learning on the ETH-UCY dataset between our method and SingularTrajectory. The ETH, HOTEL, UNIV, ZARA1, and ZARA2 scenes are denoted as A, B, C, D, and E, respectively.*

| Few-Shot           |  ETH      |  HOTEL    |  UNIV     |  ZARA1    |  ZARA2    |  AVG      |
|:------------------:|:---------:|:---------:|:---------:|:---------:|:---------:|:---------:|
| SingularTrajectory | 0.35/0.46 | 0.14/0.21 | 0.26/0.44 | 0.21/0.36 | 0.18/0.31 | 0.23/0.35 |
| Ours(TrajCLIP)     | 0.34/0.42 | 0.15/0.23 | 0.24/0.39 | 0.21/0.34 | 0.17/0.30 | 0.22/0.34 |

*Tab 3. Comparison of ADE/FDE for few-shot learning on ETH-UCY with SingularTrajectory.*

|                   |    ETH    |   HOTEL   |   UNIV    |   ZARA1   |   ZARA2   |    AVG    |
|:-----------------:|:---------:|:---------:|:---------:|:---------:|:---------:|:---------:|
| LMTraj-SUP        | 0.41/0.51 | 0.12/0.16 | 0.22/0.34 | 0.20/0.32 | 0.17/0.27 | 0.22/0.32 |
| MSN-SC            | 0.27/0.39 | 0.13/0.18 | 0.22/0.45 | 0.18/0.34 | 0.15/0.27 | 0.19/0.33 |
| HighGraph   | 0.33/0.56 | 0.13/0.21 | 0.23/0.47 | 0.19/0.33 | 0.15/0.25 | 0.21/0.36 |
| SingularTrajectory| 0.35/0.42 | 0.13/0.19 | 0.25/0.44 | 0.19/0.32 | 0.15/0.25 | 0.21/0.32 |
| Ours(TrajCLIP)    | 0.36/0.57 | 0.10/0.17 | 0.19/0.41 | 0.16/0.28 | 0.11/0.20 | 0.18/0.33 |

*Tab 4. Comparison of ADE/FDE for performance with SOTA methods on the ETH-UCY dataset.*



[1] Bae, Inhwan, Young-Jae Park, and Hae-Gon Jeon. "SingularTrajectory: Universal Trajectory Predictor Using Diffusion Model." In Proceedings of the IEEE/CVF Conference on Computer Vision and Pattern Recognition, pp. 17890-17901. 2024.

[2] Bae, Inhwan, Junoh Lee, and Hae-Gon Jeon. "Can Language Beat Numerical Regression? Language-Based Multimodal Trajectory Prediction." Proceedings of the IEEE/CVF Conference on Computer Vision and Pattern Recognition. 2024.

[3] Wong, Conghao, et al. "SocialCircle: Learning the Angle-based Social Interaction Representation for Pedestrian Trajectory Prediction." Proceedings of the IEEE/CVF Conference on Computer Vision and Pattern Recognition. 2024.

[4] Kim, Sungjune, et al. "Higher-order Relational Reasoning for Pedestrian Trajectory Prediction." Proceedings of the IEEE/CVF Conference on Computer Vision and Pattern Recognition. 2024.

---

### Decision · Program_Chairs · 2024-09-25

**Decision:**

Accept (poster)

**Comment:**

The paper presents TrajCLIP, a novel approach to pedestrian trajectory prediction using contrastive learning and idempotent networks. The method shows promise in addressing the challenges of modeling complex trajectory distributions and demonstrates strong performance across various datasets and tasks.

Initially, the paper received mixed concerns especially in terms of (1) limited discussion of limitations, (2) computational complexity, and (3) implementation details. After the rebuttal stage, most of the reviewers' concerns have been addressed and Reviewer iNsW raised his score to weak accept.

Given the novelty of the approach, strong empirical results, and the authors' thorough response to reviewers' concerns, this paper appears to make a significant contribution to the field of pedestrian trajectory prediction. The AC would like to recommend the paper for acceptance. Congratulations! Please be aware that the authors should integrate the response in the rebuttal and adequately address the reviewers' concerns in the final camera-ready version.